# Companion Animals as a Key to Success for Translating Radiation Therapy Research into the Clinic

**DOI:** 10.3390/cancers15133377

**Published:** 2023-06-27

**Authors:** Isabelle F. Vanhaezebrouck, Matthew L. Scarpelli

**Affiliations:** 1College of Veterinary Medicine, Purdue University, 625 Harrison Street, West Lafayette, IN 47907, USA; 2School of Health Sciences, Purdue University, 550 W Stadium Ave, West Lafayette, IN 47907, USA; mscarpel@purdue.edu

**Keywords:** companion animals, radiation oncology, translational research, past, present, and future

## Abstract

**Simple Summary:**

Companion animals with cancer have participated in veterinary radiation oncology research. This research has led to greater understanding of radiation safety and toxicities. However, the authors predict an even greater contribution from companion animals in the future. For the past forty years, technology in veterinary radiation oncology has progressed tremendously. Moreover, education and organization within the veterinary field have become more similar to human radiation oncology. These developments enable a greater contribution to translational research. In addition, a great advantage of veterinary research is that companion animals present with spontaneous cancers, including tumors with genetic heterogeneity and intact immune systems, making them similar to human cancers. The authors predict this advantage, along with the recent evolution in the veterinary field, will make companion animals key contributors to future discoveries in radiation oncology.

**Abstract:**

Many successful preclinical findings fail to be replicated during translation to human studies. This leads to significant resources being spent on large clinical trials, and in some cases, promising therapeutics not being pursued due to the high costs of clinical translation. These translational failures emphasize the need for improved preclinical models of human cancer so that there is a higher probability of successful clinical translation. Companion-animal cancers offer a potential solution. These cancers are more similar to human cancer than other preclinical models, with a natural evolution over time, genetic alterations, intact immune system, and a permanent adaptation to the microenvironment. These advantages have led pioneers in veterinary radiation oncology to aid human medicine by elucidating basic principles of radiation biology. More recently, the veterinary and human radiation oncology fields have increasingly collaborated to achieve advancements in education, radiotherapy techniques, and trial networks. This review describes these advancements, including significant prior research findings and the evolution of the veterinary radiation oncology discipline. It concludes by describing how companion-animal models can help shape the future of human radiotherapy. Taken as a whole, this review suggests companion-animal cancers may become widely used for preclinical radiotherapy research.

## 1. Contribution of Small Animal Companions as a Preclinical Research Model in Radiation Oncology

### 1.1. Companion Animals’ Contribution to Radiobiology Research

Companion dogs have served the field of radiobiology since veterinary academic centers were provided with radiation equipment, including orthovoltage X-rays, cobalt therapy, and linear accelerators (LINACs). This includes research by Dr. Gillette and Dr. Powers from Colorado State University (CSU), who contributed to National Health Institute (NIH)-funded radiation toxicity studies. These researchers provided knowledge regarding radiation dose effects on multiple organs, such as the brain, spinal cord, aorta, heart, trachea, and eye. This was achieved by assessing variable radiotherapy regimens (varying both fractionation and total dose) in laboratory dogs (beagles) [1,2,3,4,5]. Drs. Gillette and Powers analyzed late toxicity with pathology support, and their results included iso-effect curves and alpha-beta ratio values for different organs. Their study on the volume effect in the irradiated canine spinal cord [6] has contributed to a better understanding of the risk associated with increasing volume or dose in human patients, providing the foundation to design current mathematical models on normal-tissue complication probability (NTCP) [7,8].

Canines have also been used in toxicity studies to investigate total body irradiation. One prominent example includes previous [9] investigation of changes in total doses or dose rates in canines. This study hypothesized that fractionation reduces organ toxicity. Their findings indicated that for canines, the lethal dose for the whole body is close to 400 cGy, and there is no difference in survival based on fractionation protocols at 100 cGy × 4 fractions. In addition, most canines recovered from bone marrow ablation at 400 cGy when treated with granulocyte-stimulating factor. These studies helped inform future studies on human patients enrolled in bone marrow transplantation programs.

In the mid-1980s, intraoperative radiotherapy with electrons was proposed as a promising technique for improving tumor control. For this technique, when an external beam course has achieved its limit on toxicity, an intraoperative radiotherapy boost is performed to increase the tumor control. Often, patients would undergo surgery several weeks after an external beam radiation course. The surgery team first removes the primary cancer, and then the patient receives an intraoperative radiotherapy boost. The radiation oncology community was excited about this promising new technique, but was unsure of the risks imposed on the surrounding organs and vascular structures. There was concern these risks to normal tissues would compromise the success of reconstructive surgical techniques. To investigate these potential risks, the NIH sponsored two teams (Dr. Sindelar at NIH and Drs. Gillette and Powers at CSU) to perform toxicity studies on beagles to test different locations and doses for the boost. The study endpoint was late toxicity observation supported by pathology analysis [10,11,12,13,14,15,16,17,18,19,20,21]. The tested intraoperative radiation therapy (IORT) doses for the boost were 20, 30, and 40 Gy, and the energy (9 MeV versus 12 MeV electrons) depended on the treatment depth. Dr. Sindelar concluded, “Although the radiotolerance of differing tissues can vary among species, sufficient clinical experience has accumulated to validate the canine tissue tolerance model as representative of human tissue responses to IORT”. These researchers suggested that accumulated clinical evidence validates the canine tissue tolerance model as representative of human tissue responses to IORT [15]. One application of this technique is the intraoperative boost on bladder cancer. The sensitivity of the bladder trigone determines the dose limit, in which the bladder body tolerates irradiation better than the ureters or urethra, and a safe amount of 20 Gy has been established as an IORT boost on humans based on these preliminary canine studies. In addition, for surgical reconstruction after tumor ablation and intraoperative radiotherapy, canine studies have demonstrated that large vessels, such as the aorta, are more sensitive [11,12,17]. The risk associated with a single elevated dose (>40 Gy) are wall weakness, fragilization, and possible rupture associated with a thin vessel structure. However, medium-size vessels potentially used for reconstructive surgical techniques with a flap tolerate higher doses without significant modification on pathology exams [22].

More recently (2014), canines have contributed to kidney dose tolerance assessment with stereotactic body radiotherapy (SBRT) [23]. Healthy canines were irradiated with a total dose of 30 Gy delivered in two fractions over three days. The average kidney volume was 50–65 cc, and the irradiated volume varied from 6 to 25 cc for different locations, including the anterior kidney pole, posterior, and hilum. Blood work, imaging with scintigraphy, pyelogram, and pathology was conducted at several time points (1, 3, 6, and 12 months). The authors reported severe changes within the treated area, but minimal or no changes outside the treatment area.

Another area for collaboration between human and veterinary radiation oncologists is development of radioprotectants. For head and neck irradiation, mucositis and salivary gland atrophy constitute a significant limit in dose escalation. Amifostine (also known as WR-2721) is a sulfur compound that has the potential to limit severe mucositis (protection factor of 1.7) [24]. Canines have contributed to demonstrating the protectant role of amifostine during irradiation [25]. Canines with soft-tissue sarcoma have been utilized to investigate possible negative impacts of amifostine on tumor control with a low total dose (less than 52 Gy). However, the clinical application of amifostine has been limited due to its toxicity, resulting in hypersalivation, nausea, emesis, and hypotension. The current research focuses on the local administration of amifostine (gel) [26]. Canine patients treated for nasal tumors with intensity-modulated radiation therapy (IMRT) have been treated with another potential radioprotectant—topical Smad7. This product reduced the severity and duration of grade 3 oral mucositis, as indicated by a lower inflammatory profile (interleukin (IL)-1β, transforming growth factor (TGF)-β, and tumor necrosis factor (TNF)-α) in canine patients [27].

Radiation biologists have attempted to improve the radiation-induced cell killing during treatment by combining radiation with drugs (radiation sensitization) or other techniques. Several collaborations between veterinary schools and medical universities have provided a better understanding of the synergistic effect of hyperthermia and radiation on companion animals for soft-tissue sarcoma [28,29]. Hypoxic cell sensitizers, such as etanidazole, nitroimidazole, and tirapazamine, overcome tumor hypoxia to achieve better cell killing [30]. These findings highlight how the field of radiobiology has benefited from cooperation between veterinarians and physicians.

### 1.2. Companion Animals’ Contribution to Radiotherapy Technical Development

In addition to radiobiology discovery, companion animals have contributed to the development of radiotherapy techniques. In 1999, a safety study for an implanted balloon device, called GliaSite, was conducted on 23 canines [31]. The balloon was filled with the iodine-125 radionuclide. In this study, three canines had a small tissue resection of the frontal lobe. The other canines received either surgical resection with GliaSite implantation (eight canines) to assess baseline levels of intracranial pressure or surgical resection, GliaSite implantation, and brachytherapy treatment with 8–125 Mci of iodine-125 injected in the device (an equivalent dose of 57–63 Gy at 9 mm depth for four canines). The balloon was well tolerated and not responsible for intracranial pressure elevation. The radiation delivery was safe, and pathology analysis on the brain at the study endpoint did not raise concerns for brain toxicity. This canine study prompted the enrollment of human patients (phase I clinical trial) for a similar study. In 2002, a similar type of study was conducted with application to esophageal cancer. This study validated a stent coated with the holmium-166 radionuclide in canines, demonstrating both safety and efficacy [32]. The licensing approval of biliary stents and liver treatment with yttrium-90 has also benefited from preclinical studies in canines [33].

Additional contributions from companion animals include validation of new technology used in external beam radiotherapy. In 2001, the Henry Ford Hospital demonstrated the efficacy of high-frequency jet ventilation on canines by tracking the motion of liver-implanted fiducials in canines [34]. The movement was less than 3 mm in all directions versus an anterior–posterior range of 1.2 cm with spontaneous ventilation. This technique is essential when placing an ablative radiation dose (SBRT) with moving targets. In 2008, the precision of a tracking device for radiotherapy (Calypso^®^ medical positioning device from Varian Medical^®^) was tested at the Washington School of Medicine [35]. The initial experiences confirmed precision and accuracy based on phantom analysis. Three canines were anesthetized for the implantation of three beacons in different bronchi under fluoroscopy guidance. The navigation system localized the fiducials, and the results were validated by comparison to a three-dimensional (3D) location with onboard imaging. Sub-millimetric positioning will be confirmed before future utilization on human patients.

Canine companion studies have also contributed to development of image-guided radiotherapy. This includes utilization of dynamic contrast-enhanced magnetic resonance (DCEMR) or positron emission tomography (PET) scans with hypoxic markers. These modalities allow for spatial and temporal visualization of the hypoxic tumor area. This is beneficial for radiotherapy planning, as hypoxia causes radiotherapy resistance. Canine patients suffering from nasal tumors (adenocarcinoma and sarcoma) represent an excellent preclinical model to evaluate the benefit of these imaging techniques. For example, the University of Wisconsin Veterinary School has evaluated hypoxia and hyperproliferation using PET in canine patients, with the intent of future dose escalation in these hypoxic areas with IMRT [36,37]. In addition, Dr. Bentzen’s collaborators, affiliated with the University of Wisconsin Medical School, have worked at assessing the value of those imaging biomarkers pre- and post-radiation treatments with regression analysis models [38].

## 2. The Evolution of the Veterinary Radiation Oncology Specialty: From Education to Organized Radiotherapy Centers

Much progress has been made in harmonizing veterinary radiation oncology training so that it is similar to training for human medicine counterparts. The American College of Veterinary Radiology in the USA (https://acvr.org (accessed on 10 June 2023)) and the European Board of Specialty-Radiation Oncology in Europe (https://ebvs.eu (accessed on 10 June 2023)) under ECVDI (European College of Veterinary Diagnostic Imaging: https://www.ecvdi.org (accessed on 10 June 2023)) and ECVIM (European College of Veterinary Internal Medicine: https://Ecvim-ca.college (accessed on 10 June 2023)) govern training and certification for the veterinary radiation oncology specialty. In 2022, the American College of Veterinary Radiology reported 129 active diplomates and 35 residents. Similar to their human medicine counterparts, veterinary radiation oncology graduates must complete a residency program. After completing one or several internships, residency candidates will often participate in matching selection (veterinary internship residency matching program). The conventional residency program lasts for three years, and alternative programs can last up to five years. Programs are accredited both in academic institutions and private cancer centers. During training, residents rotate through medical oncology, imaging, anesthesia, pathology, and neurology services. Residents also receive an education in dosimetry, and they complete treatment planning dosimetry (manual treatment calculations, 3D computerized plans, IMRT, stereotactic radiosurgery (SRS), and SBRT) before the end of their program. The residents are exposed to therapy activity for a minimum of three weeks, including patient setup, machine warmup, and treatment machine delivery. Most of the residents will learn about quality assurance for IMRT and SRS plans in liaison with their affiliated physicist. In addition to this hands-on training, residents complete formal education for radiation biology, veterinary radiation technical practice, and patient clinical follow-up. For this education, the residents attend similar courses to their resident colleagues from human medical centers or journal clubs. Some veterinary radiation residents have also completed a residency in medical physics or received a PhD in radiation biology, cancer biology, or immunology to maximize their career in academia. After being trained in a residency program, veterinary radiation oncologists must be validated by a board. Thus, veterinary radiation oncology education is similar to that of colleagues in human medicine.

The Veterinary Cancer Society reports more than 105 veterinary radiation facilities (http://vetcancersociety.org (accessed on 10 June 2023)). Moreover, the field is growing fast in previously unequipped countries (Europe, the UK, and Australia), and most private veterinary cancer centers in the USA have recently upgraded their practice with a radiation oncology service. Some of this progress in veterinary radiation oncology is attributed to equipment modernization. After 2000, some veterinary radiation centers acquired IMRT capacity. Now, most of the facilities recently surveyed by the ACVR have IMRT capacity. Radiosurgery has become a standard of practice regarding the limitation of anesthesia events and hospitalization duration. Brachytherapy practice is uncommon in the USA, but more available in Europe (horses and small animals), except for strontium-90 (available in 50% of the centers in the US). Most veterinary radiation centers can provide electron beam treatments, and most radiation centers have 3D onboard imaging and radiosurgery capacity. Most facilities have a contracted part-time physicist, and a few academic centers have permanent radiation physicists. Therefore, the level of equipment and expertise is similar to what is utilized in human medicine. This greatly increases the range of cases treatable in veterinary radiation oncology.

In congruence with the educational and technical progress mentioned above, the veterinary radiation oncology community has increased its standard of practice. A group affiliated with the ACVR Radiation Oncology, Veterinary Radiation Therapy Oncology Group (VRTOG) (https://acvr.org/dashboard/diplomates/resources/veterinary-radiation-therapy-oncology-group-vrtog/ (accessed on 10 June 2023)) has worked to establish a grid for toxicity events associated with radiation, including acute and late side effects similar to the human classification (Radiation Therapy Oncology Group; RTOG) (Table 1 and Table 2). In 2023, this group will change the classification table regarding practice evolution with radiosurgery techniques. In addition, a bridge exists between the VRTOG and the American Association of Physicists in Medicine (AAPM). This is led by the AAPM Working Group for the VRTOG (https://www.aapm.org/org/structure/?committee_code=WGVRTOG (accessed on 10 June 2023)), which harmonizes veterinary radiation practice with human standards. At the last AAPM-WGVRTOG meeting, physicists volunteered to engineer a canine head phantom with diodes. This will be sent to veterinary radiation oncology facilities for physics accreditation and enrollment for sponsored studies. These medical practice improvements will increase the quality of care delivered to veterinary patients.

The cancer Moonshot^SM^ program was initiated in 2016 [39] with $1.8 billion in USA government funds to accelerate progress in cancer from prevention to screening and treatment, and a new support initiative occurred in 2022. The Center for Cancer Research (CCR) at the National Cancer Institute (NCI) now has a dedicated program for the analysis of naturally developing cancer in animals. The targeted cancers in canine companions include brain tumors, osteosarcoma, lymphoma, melanoma, primary lung cancers, and soft-tissue sarcoma. The NCI Comparative Oncology Trials Consortium (COCT) is comprised of 22 veterinary academic centers selected by the NIH-NCI. Under the leadership of the NCI-NIH, these centers will conduct research on companion cancer diseases (preclinical studies), which may be of value for developing phase I or II clinical trials in humans. This program has completed 14 trials and has three open studies. Dr. Amy Leblanc, Director of the Comparative Oncology Program, has initiated several studies on osteosarcoma and glioma and published several reviews highlighting the importance of research on companion animals for improving the cancer care of human patients [40,41]. Dr. Chand Khanna, a veterinarian and a past senior leader at the NCI, has significantly contributed to understanding the biology of human osteosarcoma and its metastasis, and he has pledged to advance progress in human research toward a strong collaboration with veterinarian oncology consortiums [42,43]. Aside from the government programs, the veterinary oncology field has received significant funding from private foundations, such as the Morris Animal Foundation and the American Kennel Club Canine Health Foundation. Thus, veterinary oncology is well positioned to contribute to future discoveries in radiotherapy research.

## 3. Future Research Contributions from Companion Animals and Veterinary Radiation Oncology

Despite progress in developing radiation therapy techniques, there are still significant side effects from radiation therapy that lower a patient’s quality of life. In addition, for some cancers, gains in clinical outcomes have been limited. This indicates additional research could be valuable to further improve radiation therapy strategies. For local disease control, there is a need for further technology development to deliver high radiation doses while limiting normal tissue toxicity. This could include, for example, future applications of flash radiation therapy or spatial fractionation (lattice, microbeam radiation therapy). These techniques provide some hope for treatment of highly radiation-resistant tumors, such as glioblastoma. In addition, therapeutic combinations such as radiation therapy and immunotherapy have shown potential to improve outcomes for metastatic cancer patients. Pioneering these techniques in companion-animal cancers could optimize future usage and successful translation in human patients (Table 3).

### 3.1. Radiotherapy and Immunotherapy Combinations

Modern equipment and advanced education in physics, radiation biology, and practice have guided veterinary radiation in its research and collaboration with human hospitals. One area of future collaboration is the development of radiotherapy and immunotherapy combinations. For radiation and immunology, the impact of dose fractionation on the microenvironment and the possibility of inducing an effect similar to vaccination in situ is shifting current radiotherapy practice [44]. Radiation immunologists have explained the activation of the native immune system with a large fraction dose of radiation and the potential to induce a systemic response called the abscopal effect [45]. In rodents, Demaria et al. [46] demonstrated that three consecutive fractions of 8–12 Gy are more beneficial than one large dose of 20 Gy. A single fraction of 20 Gy is responsible for the stimulation of the TREX enzyme, which inhibits STING, an essential pathway for the production of interferon 1 (IFN1). These effects may be amplified by combining radiotherapy with immunostimulants, such as immune checkpoint inhibitors. Despite these promising findings in rodent studies, in humans, radiotherapy-induced abscopal effects are rarely observed.

Canines represent more robust models for preclinical studies than rodents and may offer a promising avenue for successful translation of the abscopal effect and radiotherapy–immunotherapy combinations. This is because canines develop spontaneous cancers and the canine immune system acts similarly to the human immune system. The study of combining immunotherapy and radiation in the veterinary world is an active field. All the studies combining immunology and veterinary oncology as a translational model and sponsored by the NCI are presented on the website (https://www.precinctnetwork.org (accessed on 10 June 2023)).

In a pilot study, researchers at the University of California, Davis (UC Davis) have recently used the combination of a Toll-like receptor 9 (TLR9) agonist, indoleamine 2,3-dioxygenase (IDO) inhibitors, and radiation therapy (8 Gy weekly for a total of 32 Gy) on the primary site for canine patients presenting with diffuse oral melanoma (four patients) or sarcoma (one patient). These researchers observed the regression of primary cancer, the disappearance of a metastatic lesion in one canine patient, and the diminution of metastatic size in two other canine patients [47]. The same institution actively collaborates with human medicine colleagues to study dog osteosarcoma as a relevant preclinical model for combining immunotherapy and radiation. This collaboration recently resulted in a publication on the activation of natural killer cells with recombinant IL2 (rIL2) after expansion with feeder cells K562C9IL21 ex vivo [48]. This team reported a systemic immune response when treating canine osteosarcoma patients with a hypofractionated course of radiation on the primary site (9 Gy × 4 times) followed by two cycles of injections of activated natural killer cells in the primary tumor.

A Health Brain Consortium under the lead of the University of Alabama has recently studied the potential to use a genetically modified herpes simplex virus (MO32) expressing IL12 in canines presenting glioma as a preclinical model for human glioblastoma. This consortium has reported that the viral dose can be safely escalated. The viral load could be used as a single immunostimulant or in combination with IDO inhibitors. Combination treatments, including radiation therapy, could represent the next study phase [49].

Another therapeutic technique that could benefit from combination with radiation therapy is being developed at CSU for treating dog glioma. This therapy utilizes a stem cell vaccine combined with microenvironment reprogramming targeting macrophages and myeloid cells [50]. CSU has also recently studied the immune landscape of canine oral squamous cell carcinoma to combine radiation therapy and immunotherapy in canines. Future studies may contribute to developing novel therapies in human patients [51].

One hope in the veterinary world is the future commercialization of canine immune checkpoint inhibitors. These checkpoint inhibitors have shown promising results when combined with radiotherapy in rodent and human cancers. To date, however, veterinarians cannot use specific canine immune checkpoint inhibitors, such as anti-CLA4, anti-PD1, or anti-PDL1 treatments, in their daily oncology practice, but intense work in the field could resolve this limitation.

### 3.2. Targeted Radionuclide Therapy

Nuclear medicine has made developmental progress in targeted radionuclide therapy, such as using an antibody combined with radionuclides to specifically kill cancer metastasis or niches. For example, the University of Wisconsin has used alkyl phosphocholine (NM600) coated with yttrium-90 at low doses to specifically target cancer cells within the microenvironment. These studies involved mice and canines with melanoma and osteosarcoma [52]. With specific targeting, this type of nuclear medicine involves less impact on bone marrow and the risk of cell depletion. These preclinical models serve as future study designs by combining targeted radionuclides with immune checkpoint inhibitors and external beam radiation therapy in human patients. In addition, the University of Missouri has started to test a humanized antibody against GD2 ganglioside coated with Ln111 using canine osteosarcoma cell lines with positive cross-linking [53].

### 3.3. Flash Radiotherapy

In 2014, Favaudon and colleagues [54] reported a biological effect called the flash effect, which included enhanced normal tissue protection under an ultrahigh radiation dose rate (>40 Gy per second). Most conventional clinical machines deliver a fraction in one or several minutes. Flash radiation machines produce the same dose in less than a millisecond or microsecond. Understanding this biological effect is complex. The flash effect has been observed for several organs, such as the lung, skin, and brain. There is current enthusiasm for this biological effect in the radiation community as this discovery may transform radiation practice. It would allow radiation oncologists to place a higher dose in the tumor while preserving the normal tissues, increasing the therapeutic ratio. Montay-Gruel and colleagues demonstrated the preservation of cognitive function while irradiating full-brain mice at a dose rate allowing the flash effect [55]. Limoli and colleagues described multifactorial causes to explain the flash effect [56]. Many leading institutions are currently trying to obtain flash irradiators, and most machines currently use electron beams for flash treatments. However, the usage of these machines remains mainly at the research level. One patient has been treated at the CHV Lausanne, Switzerland for recurrent T cell lymphoma using flash radiation [57], and clinical trials on patients are expected to be performed.

Human radiation institutions are collaborating with veterinary centers for a preclinical phase using flash radiation, including studies on canines, cats, and pigs. In cats affected by nasal squamous cell carcinoma, Vozenin and colleagues demonstrated the sparing of skin with flash radiation with [57] a maximum dose of 25 Gy in one fraction. However, for another trial with a dose escalation of 30 Gy, unacceptable maxillary toxicity (necrosis) led to premature study termination [58]. Further investigations are necessary to determine the optimal dose distribution, dose prescription, and dose rate for these flash radiation techniques.

Researchers at Oxford University, UK have collaborated with the veterinary school in Denmark for a dose escalation study using flash electrons on canines affected by a superficial tumor. These researchers used a transformed LINAC to produce ultrahigh-dose-rate electrons. Fraction size was progressively increased from 30 to 35 Gy without any significant side effects, except for one patient suffering from grade 3 skin toxicity [59].

While flash electrons are a promising technique for treating superficial tumors, their limitation is treatment depth. The first machine generations were transformed LINACs followed by specially engineered flash irradiators at 6 MeV. The industry currently develops devices with 9–10 MeV capacity, but radiation oncologists expect modified proton machines to obtain the capacity to treat at greater depths. As pioneers in this technology, researchers at the University of Pennsylvania (U Penn) have studied the efficacy and toxicity of both types of delivery, namely, proton and flash proton therapy, on mice injected with orthoptic sarcoma using the Proteus Plus-230 MeV Cyclotron irradiator from IBA (1348, Louvain-La-Neuve, Belgium) [60]. Flash therapy was reported to have less toxicity according to clinical exams and pathology. Researchers at U Penn have also enrolled 20 canine patients with primary osteosarcoma, which were equally allocated to receive either proton therapy or flash proton therapy (8 or 12 Gy single fraction) to treat the primary osteosarcoma. Irradiation was delivered within a 2.5 cm circle, including a part of the tumor and a part of the healthy tissue. Five days after irradiation, the leg was amputated and sent to pathology to confirm the protective effect of flash therapy on irradiated normal tissue by quantifying TGFβ production, and the results indicated lower TGFβ production in the flash therapy group [60].

### 3.4. Spatially Fractionated Radiotherapy

Spatial fractionation is another active area of research. One possible way to overcome radiation toxicity is to deliver non-uniform radiation doses within the tumor. Conventionally, this includes high doses and valley radiation doses (minimal doses). Both bystander and abscopal effects have been implicated regarding how the tumor cells within the dose valley are effectively treated. One form of spatial fractionation, lattice radiation therapy, has progressed from using a two-dimensional (2D) grid to 3D models using volumetric modulated arc therapy (VMAT) plans [61]. A new technology, microbeam radiation therapy, is produced by synchrotrons and could transform this research area. A pilot study was performed in a synchrotron consortium facility in Grenoble, France, on a French bulldog presenting with a glioma [62]. Five coplanar microbeams were selected to treat the patient (width was 50 microns with a pitch of 200–400 microns, and the dose within the valley was 5–7% of the peak dose). On the dosimetry aspect, there was considerable dose heterogeneity within the tumor. After treatment, the patient improved in terms of seizure control. Serial magnetic resonance imaging (MRI) over three months showed a progressive tumor reduction of 87.4% and the absence of brain edema. Some entrance doses registered peaked at 60 Gy. The next step for the study team is to improve the correlation between the measurement of doses delivered and doses prescribed because a 25% gap was observed. They expect to enroll 27 canines before 2025 for a dose-escalation study and to gather additional data on treatment efficacy and toxicity (short-term and long-term).

### 3.5. Limitations

As with all preclinical models, companion animals have their limitations. First is the research technology availability. Cutting-edge radiation therapy techniques often require companion animals to travel long distances based on limited availability (e.g., flash protons, synchrotrons, etc.), so only motivated owners will accept enrollment. Furthermore, the preclinical trial relies on the owner’s acceptance and dedication to medical procedures on their pet, such as multiple biopsies, imaging, and repetitive anesthesia for biology investigation. Owners have to respect a strict agenda for the medical follow-up and this may preclude some pets from participation. In addition, some drugs are limited in veterinary medicine, such as commercial antibodies, including immune checkpoint inhibitors, which can limit the types of studies performed. The principal reasons are limited veterinary research and the lack of investment from leading pharmaceutical companies regarding the small size of the veterinary market.

## 4. Conclusions

The aim of the present review was to highlight the importance of the collaboration between the veterinary and human radiation oncology fields, from the first external beam machine and the development of radiobiology to the modern era with ultraperformance modern LINACs and the integration of radiation oncology and immunotherapy. Moreover, the latest developments in technology, such as the flash effect (ultrahigh-dose radiation delivery), particle radiation therapy, or spatial fractionation (including the usage of synchrotrons for microbeams), benefit from the utilization of canine patients for preclinical research for preliminary studies on efficacy, toxicity, and technique optimization. This collaboration has shaped the veterinary radiation oncology practice and places it into the modern era and precision medicine. The future of the partnership between the veterinary and human radiation oncology fields could involve veterinary radiation oncologists integrating with human cancer radiation departments at the level of preclinical research and a platform dedicated to the enrollment of veterinary patients. Large multi-institute consortiums will remove financial barriers and provide access to technology contributing to increased treatment of veterinary patients with natural cancers. In addition, human medicine will certainly benefit from data obtained from these veterinary patients, underlying the significant contribution these animals can provide in improving radiation oncology practice.

## Figures and Tables

**Table 1 cancers-15-03377-t001:** Acute effects. From Toxicity Criteria of the VRTOG group: Acute Radiation Morbidity Scoring Scheme (ACVR.org/VRTOG (accessed on 10 June 2023), courtesy of Dr. Ladue and Dr. Klein).

Organ/Tissue	Morbidity Scoring			
	0	1	2	3
Skin	no change over baseline	erythema, dry desquamation, alopecia/epilation	patchy moist desquamation without edema	confluent moist desquamation with edema and/or ulceration, necrosis, hemorrhage
Mucous membranes/oral cavity	no change over baseline	injection without mucositis	patchy mucositis with patient seemingly pain-free	confluent fibrinous mucositis necessitating analgesia, ulceration, hemorrhage, necrosis
Eye	no change over baseline	mild conjunctivitis and/or scleral injection	KCS requiring artificial tears, moderate conjunctivitis, or iritis necessitating therapy	severe keratitis with corneal ulceration and/or loss of vision, glaucoma
Ear	no change over baseline	mild external otitis with erythema, pruritis 2° to dry desquamation, not requiring therapy	moderate external otitis requiring topical medication	severe external otitis with discharge and moist desquamation
Lower GI	no change over baseline	change in quality of bowel habits not requiring medication, rectal discomfort	diarrhea requiring medication, rectal discomfort requiring analgesia	diarrhea requiring parenteral support, bloody discharge necessitating medical attention, fistula, perforation
Genitourinary	no change over baseline	change in frequency of urination not requiring medication	change in frequency of urination necessitating medication	gross hematuria or bladder obstruction
CNS	no change over baseline	minor neurologic findings not necessitating more than prednisone therapy	neurologic findings necessitating more than prednisone therapy	serious neurologic impairment such as paralysis, coma, obtunded
Lung	no change over baseline	alveolar infiltrate; cough—no treatment required	dense alveolar infiltrate; cough—treatment required	Dyspnea

**Table 2 cancers-15-03377-t002:** Late effects. From Toxicity Criteria of the VRTOG group: Late Radiation Morbidity Scoring Scheme (ACVR.org/VRTOG (accessed on 10 June 2023), courtesy of Dr.Ladue and Dr. Klein).

Organ/Tissue	Morbidity Scoring			
	0	1	2	3
Skin/hair	None	alopecia hyperpigmentation leukotrichia	asymptomatic induration (fibrosis)	severe induration causing physical impairment, necrosis
CNS	None	mild neurologic signs not necessitating more than prednisone therapy	neurologic signs necessitating more than prednisone therapy	seizures, paralysis, coma
Eye	None	asymptomatic cataracts, KCS	symptomatic cataracts, keratitis, corneal ulceration, minor retinopathy, mild to moderate glaucoma	panophthalmitis, blindness, severe glaucoma, retinal detachment
Bone	None	pain on palpation	radiographic changes	necrosis
Lung	None	patchy radiographic infiltrates	dense radiographic infiltrates	symptomatic fibrosis, pneumonitis
Heart	None	ECG changes	pericardial effusion	pericardial tamponade, congestive heart failure
Joint	None	Stiffness	decreased range of motion	complete fixation
Bladder	None	microscopic hematuria	pollakiuria, dysuria, hematuria	contracted bladder

**Table 3 cancers-15-03377-t003:** Summary of the advantages of using companion animals for preclinical research and emerging radiation techniques or combined therapy of interest.

Advantages of Companion Animals in Preclinical Research	Emerging Radiation Techniques Studied in Companion Animals
Spontaneous disease with months or years for evolution	Trials on radiation therapy combined with immunotherapy; research for the abscopal effect
Genotype and phenotype heterogeneity	Theragnostics
Some cancers are similar to their human counterpart e.g., glioma, melanoma, or osteosarcoma including metastatic potential (biology)	Ultrahigh-dose rate radiation e.g., flash
Tumor microenvironment is natural	Lattice radiation therapy
Natural life conditions with a robust immune system	Microbeam radiation therapy
Larger than rodents. This allows use of the same equipment and dosimetry as humans: CT scan, MRI, PET scan, Modern LINAC, Cyberknife, etc.	Particle therapy: protons, carbon, heavy ions
Follow-up and response to cancer therapy fast (months to a few years on average) in comparison to humans (decade)-Dogs life span ~15 years	Neutrons; boron-neutron-capture therapy
Money saving, could avoid billion dollars wasted on clinical trials	Novel isotopes and forms of brachytherapy

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
