# Peer review of "Companion Animals as a Key to Success for Translating Radiation Therapy Research into the Clinic"

_cancers, 2023, doi:10.3390/cancers15133377_

Round 1

Reviewer 1 Report

This review article is well written and organized. I especially liked the part in which the authors describe the evolution of the veterinary radiation oncology specialty. After reading the review, one end up agreeing with the authors on the fact that companion animals are the link to translation of RT research into the clinic.

Before publication I would recommend the following revisions.

Major revisions:

-I think it is necessary to add an artwork (or eventually a table) summarizing all the advantages of using companion animals and all the emerging experimental RT techniques they could be of aid to. In my opinion, this would improve the MS very much.

-At the beginning of section 3, a general short paragraph about the rationale behind the clinical need of new RT strategies should be added. In this paragraph, it should be explained why there is this need despite the massive current use of CONV RT for cancer patients.

-In section 3.3 the authors should cite and describe the following publication from last year doi: 10.1158/1078-0432.CCR-22-0262 “Dose- and Volume-Limiting Late Toxicity of FLASH Radiotherapy in Cats with Squamous Cell Carcinoma of the Nasal Planum and in Mini Pigs”. The authors should especially mention the dose limitations of FLASH RT due to the onset of late maxillary and mucosal necrosis in cats.

-In section 3.4 the authors refer to Spatially fractionated radiotherapy (SFRT) as a synonym of LATTICE RT. To the best of my knowledge, LATTICE RT is only a subtype of SFRT defined as the 3D evolution of GRID RT, where radiation is delivered in multiple divergent beams focusing their vertices on the target. This is indeed not the case for the study they mention as “Lattice radiation with synchrotron” in the same paragraph (ref 61). In this study, five coplanar and conformal MRT (multislit collimator based) fields were employed.  This should be revised.

-As all preclinical models, companion animals have their limitations. The authors should state these limitations clearly in this review. An example could be the limited availability of commercial antibodies for this species as well as limitations in obtaining frequent biopsies to be investigated for biological mechanisms. At the end of section 3.1 the authors mention the nonexistence of canine immune checkpoint inhibitors, but I feel more should be said.

Minor revisions:

-The authors should revise the word spacing in the whole text: two spaces instead of one, no space between text and ref or between dose and Gy.

-Line 324, Limoli is misspelled.

-Line 328, the authors should specify the hospital and the country where the patient was treated.

-Reference list font size should be homogenous.

Author Response

Dear reviewer ,

Thank you for your guidance and your dedicated  time,

Please consider these revisions,

Best,

the authors 

Major revisions:

-I think it is necessary to add an artwork (or eventually a table) summarizing all the advantages of using companion animals and all the emerging experimental RT techniques they could be of aid to. In my opinion, this would improve the MS very much.

Please consider  our addition of table 3

-At the beginning of section 3, a general short paragraph about the rationale behind the clinical need of new RT strategies should be added. In this paragraph, it should be explained why there is this need despite the massive current use of CONV RT for cancer patients.

A transition paragraph  has been added, with the rationale for new technologies and the potential involvement of companion animals

-In section 3.3 the authors should cite and describe the following publication from last year doi: 10.1158/1078-0432.CCR-22-0262 “Dose- and Volume-Limiting Late Toxicity of FLASH Radiotherapy in Cats with Squamous Cell Carcinoma of the Nasal Planum and in Mini Pigs”. The authors should especially mention the dose limitations of FLASH RT due to the onset of late maxillary and mucosal necrosis in cats.

We followed your recommendation to add the Carla Rohrer article

-In section 3.4 the authors refer to Spatially fractionated radiotherapy (SFRT) as a synonym of LATTICE RT. To the best of my knowledge, LATTICE RT is only a subtype of SFRT defined as the 3D evolution of GRID RT, where radiation is delivered in multiple divergent beams focusing their vertices on the target. This is indeed not the case for the study they mention as “Lattice radiation with synchrotron” in the same paragraph (ref 61). In this study, five coplanar and conformal MRT (multislit collimator based) fields were employed.  This should be revised.

We corrected this full paragraph with the hope of bringing clarification

-As all preclinical models, companion animals have their limitations. The authors should state these limitations clearly in this review. An example could be the limited availability of commercial antibodies for this species as well as limitations in obtaining frequent biopsies to be investigated for biological mechanisms. At the end of section 3.1 the authors mention the nonexistence of canine immune checkpoint inhibitors, but I feel more should be said.

I added a paragraph to cover the main limitations 

Minor revisions:

-The authors should revise the word spacing in the whole text: two spaces instead of one, no space between text and ref or between dose and Gy.

corrected

-Line 324, Limoli is misspelled.

Thank you

-Line 328, the authors should specify the hospital and the country where the patient was treated.

Corrected

-Reference list font size should be homogenous.

Corrected

Reviewer 2 Report

General comments: This is a review article highlighting the various collaborative advancements that have helped to move forward both veterinary and human radiation oncology. This is a thorough and well written paper. I have very minor edits, but would recommend a thorough grammatical and spelling check. I would also make sure you are listing references for many of these advancements. I would love to look at these references and learn more.

46: Add “whole body” to this sentence so the reader understands this reference better.

88: How can blood work show severe changes within the treated area? Can you describe this further?

109: Correct spelling for etanidazole.

168: “…the alternative programs can last up to five years”

176: remove comma after addition

176-179: It doesn’t have to be formal training. Book clubs are satisfactory for ACVR as long as there is training in these sections.

183: I would recommend removing “now”.

194: “available in 50% of the centers in the USA”

214: Reference

258: Reference for Demaria et al

264: Remove “a” or say “model”

291: This may also be a paper to consider adding to the glioma research with immunotherapy that has potential to be combined with RT. (Reprogramming the Canine Glioma Microenvironment with Tumor Vaccination plus Oral Losartan and Propranolol Induces Objective Responses by Ammons et al).

356: This is amazing work! Do you have a reference?

383: The radiation oncology group at Missouri (Charles Maitz) presented an abstract at Radiation Research Society in 2022 on lattice radiation as well. Not sure if you want to include an abstract, but he had some impressive data.

Author Response

Dear Reviewer 2,

Thank you for your guidance and your dedicated time with us.

I hope we followed  your remarks and recommendations,

best wishes,

the Authors

General comments: This is a review article highlighting the various collaborative advancements that have helped to move forward both veterinary and human radiation oncology. This is a thorough and well written paper. I have very minor edits, but would recommend a thorough grammatical and spelling check. I would also make sure you are listing references for many of these advancements. I would love to look at these references and learn more.

46: Add “whole body” to this sentence so the reader understands this reference better.

corrected  now at line 57

88: How can blood work show severe changes within the treated area? Can you describe this further?

I tried to bring some clarification, I hope it will help the reading

109: Correct spelling for etanidazole.

Corrected; thank you 

168: “…the alternative programs can last up to five years”

corrected

176: remove comma after addition

We modified slightly for lecture  facilitation 175-194

176-179: It doesn’t have to be formal training. Book clubs are satisfactory for ACVR as long as there is training in these sections.

we corrected with journal clubs

183: I would recommend removing “now”.

done; thank you

194: “available in 50% of the centers in the USA”

done; thank you

214: Reference

It was mentioned several times at the ACVR ( Steve Saparento) congress and at the AAPM meetings. I hope the added clarification will help.

258: Reference for Demaria et al

46

264: Remove “a” or say “model”

Corrected, Thank you:_)

291: This may also be a paper to consider adding to the glioma research with immunotherapy that has potential to be combined with RT. (Reprogramming the Canine Glioma Microenvironment with Tumor Vaccination plus Oral Losartan and Propranolol Induces Objective Responses by Ammons et al).

  Thank you, we have added it.

356: This is amazing work! Do you have a reference?

  1. Velalopolou A, Karagounis I, Cramer G, Kim M, Skoufos G , Goia D. Flash Proton radiotherapy spares normal epithelial and mesenchymal tissues while preserving sarcoma response. Cancer Res 2021;81;(18)4808-4821.doi:10.1158/0008-5472-CAN-21-1500

383: The radiation oncology group at Missouri (Charles Maitz) presented an abstract at Radiation Research Society in 2022 on lattice radiation as well. Not sure if you want to include an abstract, but he had some impressive data.

I could not track this one apology; will you share it when you have time?

Best wishes, and thanks again for your time.

Reviewer 3 Report

This article reviews the continuation of large companion and research animals to the field of radiobiology and radiation oncology. Overall this is interesting.

While the title states that this is about companion animals much of the work cited is done in laboratory research dogs and not client owned companion animals.  Therefore I would suggest modifying the title accordingly. 

One additional but not mentioned advantage of using dogs in this type of research is that dogs are of larger size than rodents and this allows the use of the same equipment and dosimetry as is used in humans.

I also have some specific comments below:

Abstract:

Instead of companion cancers - consider companion animal cancers.

I think the final statement may be too strong. Given the relatively small number of veterinary radiation oncologists and centers cited in this paper it is unlikely that this type of research could become the "standard".

Section 1.1 - instead of standardized canines consider calling them laboratory animal dogs.

Line 47 - you mean 100cGy/fx x 4 fractions?

In this section and throughout instead of Mev use MeV.

Section 2

you refer to the ACR - I think this might actually be the American College of Veterinary Radiology. 

For quality control do you mean quality assurance?

Do you mean 3D on board imaging not 4D?

Section 3:

Typo on line 260 - word combinging.

In lines 281 and 284, Do you mean IL-2 not IL-21?

Do you have a reference for the Penn Flash studies.

The manuscript suffers from awkward use of the English language throughout and would benefit from editing by a native speaker.

There are several typographical errors throughout the manuscript.

Author Response

Dear reviewer, Thank you so much for your time and precious guidance.

You will find those corrections

Best wishes from the Authors

This article reviews the continuation of large companion and research animals to the field of radiobiology and radiation oncology. Overall this is interesting.

While the title states that this is about companion animals much of the work cited is done in laboratory research dogs and not client owned companion animals.  Therefore I would suggest modifying the title accordingly. 

 We corrected the title with the addition -translational research.   

One additional but not mentioned advantage of using dogs in this type of research is that dogs are of larger size than rodents and this allows the use of the same equipment and dosimetry as is used in humans.

we added a table (3 ) to summarizes on the equipment

I also have some specific comments below:

Abstract:

Instead of companion cancers - consider companion animal cancers.

Corrected

I think the final statement may be too strong. Given the relatively small number of veterinary radiation oncologists and centers cited in this paper it is unlikely that this type of research could become the "standard".

We attenuated the statement:_)

Section 1.1 - instead of standardized canines consider calling them laboratory animal dogs.

Corrected

Line 47 - you mean 100cGy/fx x 4 fractions?

In this section and throughout instead of Mev use MeV. 

fixed both

Section 2

you refer to the ACR - I think this might actually be the American College of Veterinary Radiology. 

We updated ACVR, Thank you for the catch

For quality control do you mean quality assurance?

Corrected

Do you mean 3D on board imaging not 4D?

Corrected. Thank you

Section 3:

Typo on line 260 - word combinging.

The spelling is corrected. Thank you.

In lines 281 and 284, Do you mean IL-2 not IL-21?

We brought modifications for clarification ( IL 2 was applied for treatment, and IL 21 was only used to stimulate the feeders). The paragraph is new.

Do you have a reference for the Penn Flash studies?

added ( 60)

Velalopolou A, Karagounis I, Cramer G, Kim M, Skoufos G , Goia D. Flash Proton radiotherapy spares normal epithelial and mesenchymal tissues while preserving sarcoma response. Cancer Res 2021;81;(18)4808-4821.doi:10.1158/0008-5472-CAN-21-1500